# Adiponectin—Consideration for its Role in Skeletal Muscle Health

**DOI:** 10.3390/ijms20071528

**Published:** 2019-03-27

**Authors:** Matthew P. Krause, Kevin J. Milne, Thomas J. Hawke

**Affiliations:** 1Department of Kinesiology, Faculty of Human Kinetics, University of Windsor, 401 Sunset Avenue, Windsor, ON N9B 3P4, Canada; kjmilne@uwindsor.ca; 2Department of Pathology and Molecular Medicine, Faculty of Health Sciences, McMaster University, 1280 Main Street, Hamilton, ON L8S 4L8, Canada; hawke@mcmaster.ca

**Keywords:** skeletal muscle, regeneration, adiponectin isoforms, exercise, training

## Abstract

Adiponectin regulates metabolism through blood glucose control and fatty acid oxidation, partly mediated by downstream effects of adiponectin signaling in skeletal muscle. More recently, skeletal muscle has been identified as a source of adiponectin expression, fueling interest in the role of adiponectin as both a circulating adipokine and a locally expressed paracrine/autocrine factor. In addition to being metabolically responsive, skeletal muscle functional capacity, calcium handling, growth and maintenance, regenerative capacity, and susceptibility to chronic inflammation are all strongly influenced by adiponectin stimulation. Furthermore, physical exercise has clear links to adiponectin expression and circulating concentrations in healthy and diseased populations. Greater physical activity is generally related to higher adiponectin expression while lower adiponectin levels are found in inactive obese, pre-diabetic, and diabetic populations. Exercise training typically restores plasma adiponectin and is associated with improved insulin sensitivity. Thus, the role of adiponectin signaling in skeletal muscle has expanded beyond that of a metabolic regulator to include several aspects of skeletal muscle function and maintenance critical to muscle health, many of which are responsive to, and mediated by, physical exercise.

## 1. Introduction

Since the discovery of adiponectin over 20 years ago [1], nearly 20,000 scientific articles have been published on this adipokine; reflecting an intense interest from the scientific community. Although originally identified as an adipose tissue secreted protein, adiponectin is now known to be expressed by multiple tissues including skeletal muscle. In conjunction with other canonical metabolic hormones (e.g., insulin, leptin, etc.), adiponectin helps to regulate metabolism through blood glucose control and fatty acid oxidation [2,3,4,5]. Despite being expressed and secreted by adipocytes, obesity-associated metabolic disorders such as insulin resistance and type 2 diabetes (T2D) are inversely related to adiponectin levels (i.e., circulating adiponectin decreases despite greater fat mass) [5,6]. Furthermore, low adiponectin levels are related to an increased rate of progression of diabetic complications such as nephropathy, retinopathy, and cardiomyopathy [7]. Thus, much of the research focus has been on elucidating the mechanistic roles played by adiponectin in regulating metabolism across multiple tissues, and how its expression is regulated under normal and pathophysiological circumstances. More recently, other physiological roles of adiponectin have emerged, including that skeletal muscle both expresses and is sensitive to adiponectin. Consequently, the purpose of this review is to highlight the physiological roles of adiponectin in skeletal muscle and the pathophysiology related to dysregulated adiponectin expression. Given the potency of regular physical exercise to improve metabolic control, this review will also examine how adiponectin expression is altered by exercise and whether benefits of exercise are mediated, at least in part, by the actions of adiponectin. 

## 2. Expression and Post-Translational Modification of Adiponectin

Well over 200 proteins are reported to be expressed and secreted by human adipocytes, one of which is adiponectin (also referred to as adipocyte complement-related protein of 30 kDa [Acrp30], Adipocyte, C1q, and collagen domain-containing protein [ACDC], or Adipose most abundant gene transcript 1 protein [apM-1]) [8]. Originally, expression and release of adiponectin into the circulation was thought to be restricted to adipose tissue [1], however, it is now established that adiponectin is produced and secreted from a number of cell types, including skeletal and cardiac muscles [9,10,11,12,13,14,15,16]. Adiponectin is part of a large family of secreted protein hormones, the C1q TNFα Related Proteins (CTRP), many of which have overlapping biological functions [17]. At least eight isoforms of adiponectin exist following post-translational modifications of the initial gene product [18]. In the plasma, adiponectin exists as low molecular weight trimers (LMW) that can associate with one another to form middle molecular weight hexamers and high molecular weight (HMW) multimers of various sizes [19] (Figure 1), while the adiponectin monomer is not detected in the circulation. These post-translational modifications and associations impact the stability and biological activity of adiponectin in the circulation [18,19]. Indeed, HMW adiponectin has been shown to have a greater predictive power for insulin resistance than total plasma adiponectin [20]. Adiponectin is one of the most abundant adipokines in the plasma, circulating in the range of approximately 5 to 30 μg/mL with a half-life of 13 and 17.5 h for the HMW and low molecular weight isoforms, respectively [21]. This expression level is approximately 0.05% of total serum protein content. In comparison, other notable adipokines have been reported in the ng/mL scale. For example, leptin and plasminogen activator inhibitor (PAI)-1, range between 1 to 200 ng/mL [22,23] and 15 to 550 ng/mL [24], respectively. 

Through proteolytic cleavage, adiponectin can also exist as globular adiponectin (gAd; Figure 1) and reports suggest that, although it is expressed at very low levels, gAd displays biological activities that are distinct from the properties of the full-length adiponectin protein [25,26,27,28]. Throughout the remainder of the review, the isoform of adiponectin (globular, trimeric, hexameric, or HMW) will be indicated where possible. However, a major limitation in how the findings of adiponectin studies are interpreted is that the adiponectin isoform is often not delineated, possibly due to the reliance on pan-adiponectin antibodies for detection.

The secretion, stability, and signaling function/potency of adiponectin is dependent not only on multimeric conformation, but how adiponectin is post-translationally modified. Adiponectin shares structural similarities with some collagen types and, similar to collagen, is glycosylated and hydroxylated as part of its post-translational modification [18,29,30]. Trimeric (LMW) adiponectin is stabilized by interactions of the collagenous domains, while the hexameric and HMW forms further require disulfide bond formation between cysteine residues [29,30]. Quenching of available cysteine residues (through excessive fumarate causing succination of cysteine) prevents the post-translational modifications necessary to produce competent hexamers and HMW adiponectin in type 2, but not type 1, diabetic rodents [31,32,33]. Succination is a post-translational modification for many proteins and appears to be upregulated in obese and diabetic rodents in multiple tissues including skeletal muscle [33]. Consequently, it is likely that adiponectin expressed by tissues other than adipose is similarly affected by excessive fumarate. The half-life of circulating adiponectin also appears to be dependent on post-translational modification. Consistent across species [34], adiponectin has been demonstrated to be modified by the addition of sialic acid to O-linked glycans (referred to as sialylation) and the desialylation of adiponectin results in accelerated clearance of adiponectin from the circulation [35].

Adiponectin expression follows a circadian rhythm, with circulating concentrations peaking in the early afternoon [36,37], although the impact of this rhythm is not well understood. Obesity and the progression from insulin resistance to diabetes has been linked to disruptions in circadian rhythm stemming from a cycle of disrupted sleep and poor eating habits. A potential link between disrupted circadian rhythms and metabolic disease progression is the disruption of rhythmic adiponectin expression and signaling. For example, mice switched from a normal diet to a high fat diet (to induce obesity and insulin resistance) caused a phase delay and general decrease in adiponectin expression, as well as phase delays in adiponectin receptor mRNA peaks [38], similar to observations of obese, diabetic KK-A(y) mice [39]. Conversely, mice with disrupted expression of circadian rhythm regulators (Bmal1 and Clock) exhibited an increase in adiponectin expression [40,41]. Interestingly, mice that were subjected to repeated weight cycling demonstrated disrupted expression of several clock genes with no significant alteration to plasma adiponectin despite increased adiposity [42]. Clearly, this potential relationship between circadian rhythms, adiponectin expression, and metabolic diseases is of tremendous importance and requires further attention. 

## 3. Adiponectin Effects in Skeletal Muscle

### 3.1. Muscle Function and Calcium Handling

There is little evidence of a direct relationship between adiponectin and skeletal muscle contractile capacity, and the studies inferring such a relationship are limited. While adiponectin KO mice displayed a reduction in peak force [13], adiponectin receptor 1 (AdipoR1) KO mice displayed poor capacity for endurance exercise and a decreased type I fiber percentage but were not tested for peak force [43]. In contrast, a study of young and elderly BMI- and physical activity habit-matched males and females reported no correlation between adiponectin levels and contractile force output [44]. 

Despite scattered evidence of an effect on contractile force, adiponectin does appear to regulate intramyocellular calcium concentration; important in dictating the contractile force output in muscle. For example, adding adiponectin to the culture media of differentiated C2C12 myotubes resulted in a rapid increase in intracellular calcium, an effect that is abolished by siRNA knockdown of AdipoR1 [43], while a similar effect is also observed in C2C12 myoblasts [45]. These studies offer evidence that the adiponectin-mediated calcium influx is mediated both by calcium from sarcoplasmic reticulum stores and the extracellular space [43,45]. Given that intramyocellular calcium modulates contractile force output, myosin light chain phosphorylation state, and a multitude of gene expression responses [46], adiponectin likely plays a role in calcium-mediated events in skeletal muscle, assuming that cellular observations translate *in vivo*. Indeed, an adiponectin-induced increase in myocellular calcium has been linked to activation of calmodulin-kinase activation and transcription of PGC-1α [43,45]. Further, adiponectin has recently been shown to influence calcium transients in cardiomyocytes through the regulation of sarcoplasmic reticulum calcium ATPase (SERCA) function [47], thereby presenting another method by which adiponectin may be linked to contractile function through calcium handling.

In both human and animal models of diabetes, reduced skeletal muscle contractile capacity is typically observed, however, a unified mechanism for this reduction remains elusive [13,48,49,50]. A recent study using a high-fat diet (HFD) rat model to induce diabetes (but also characterized by low adiponectin expression) found reduced peak twitch and tetanic force and a prolonged half-relaxation time, in addition to reduced SERCA gene expression in the gastrocnemius [51]. However, HFD rats treated with adiponectin transfection in one gastrocnemius saw partial restoration of force production, attributable to the restoration of SERCA expression. Further, exercise training had a similar effect on restoring SERCA expression and contractile parameters, although it is noteworthy that adiponectin transfection in combination with exercise training did not have a synergistic effect [51]. The observation of reduced muscle function is in agreement with previous studies on the effect of a HFD [49] or adiponectin-KO [13]. Consequently, we speculate that adiponectin has limited acute effects on muscle contraction, but that chronic muscle adiponectin signaling, or lack thereof, in diabetic or adiponectin KO models leads to changes in calcium handling, and thus influences contractile capacity via both calcium availability and changes in gene expression.

### 3.2. Muscle Development, Growth, Maintenance, and Aging

Adiponectin appears to play a role in regulating muscle mass, with recent mechanistic studies demonstrating it as a critical signal for muscle regeneration and suppression of proteolysis [25,52,53,54,55,56,57,58]. Epidemiological studies support the idea that adiponectin aids in the development and maintenance of muscle mass. For example, adiponectin was recently implicated in a study of adolescent idiopathic scoliosis (AIS), a common form of spinal deformity [59]. It is thought that unequal bilateral development of the paravertebral muscles leads to the development of lateral curvatures of the spine. Muscle samples of paravertebral muscles from the concave (more developed) and convex sides of AIS were analyzed via RNAseq. Interestingly, among other genes, adiponectin expression was found to be high on the concave side relative to the convex side [59], suggesting that this imbalance is related to unequal rates of paravertebral development. 

Similarly, there is evidence that adiponectin provides a protective effect in muscle wasting conditions. Muscle wasting in sarcopenia is associated with aging and is driven by multiple factors including motor neuron degeneration and hormonal changes. Adiponectin was found to be significantly decreased in sarcopenic compared to non-sarcopenic adults [60]. However, in another study, young and elderly (non-sarcopenic) participants matched for physical activity habits demonstrated no difference in muscle mass or circulating adiponectin levels [44]. It is worth noting that in a study of young vs old mice, adiponectin expression was markedly higher in old EDL muscle compared to young, but AdipoR2 was not expressed as highly in old compared to young muscle [61], suggesting that disrupted adiponectin signaling, rather than adiponectin levels, may be problematic in some cases. 

Together, these finding are surprisingly at odds with other studies suggesting that higher adiponectin levels drive muscle wasting. Adiponectin levels were found to be significantly elevated in sarcopenic males with cardiovascular disease (CVD) compared to non-sarcopenic, CVD controls [62]. Furthermore, adiponectin levels negatively correlated with functional measures such as grip strength and gait speed [62]. A similar negative relationship between adiponectin and muscle function has been demonstrated in other studies examining middle aged and elderly people with and without CVD [63,64,65]. As well, in a study of spinal and bulbar muscular atrophy patients, circulating adiponectin levels were found to be higher compared to age-matched healthy control participants, although circulating adiponectin levels did not significantly correlate with a composite muscle function score [66]. These epidemiological studies are supported by an *in vitro* study that manipulated adiponectin signaling with the use of AdipoRon [61], a small molecule agonist of AdipoR1 and R2 [67]. AdipoRon treatment reduced protein content and newly-formed myotube size in C2C12 cells, while reducing muscle fiber size in mouse plantaris muscle [61]. Given the well-defined role of adiponectin as an activator of adenosine monophosphate-activated protein kinase (AMPK) [4,68] and AMPK activity inhibits the mammalian target of rapamycin (mTOR) [69], perhaps it should not be surprising that elevated adiponectin signaling would negatively correlate with muscle mass/function. We speculate that there is a certain healthy range of adiponectin concentrations and/or signaling and significant deviations below or above that range is pathological. Further study is required to resolve these apparently opposing roles of adiponectin in the regulation of muscle mass in health and various disease states.

### 3.3. Skeletal Muscle Regeneration and Adaptive Capacity

Early studies by Fiaschi et al. provided evidence for the impact of adiponectin on skeletal muscle regeneration. This group first reported that proliferating skeletal muscle cells responded to the globular isoform of adiponectin by exiting the cell cycle, committing to the myogenic lineage, and driving differentiation [52]. This response appeared to be mediated through redox signaling since treatment with the ROS scavenger, N-acetyl cysteine (NAC), blunted the adiponectin-induced muscle differentiation [52]. A follow-up study demonstrated that satellite cells isolated from murine tibialis anterior muscles were sensitive to both full-length and globular adiponectin, though the latter induced a greater motility in satellite cells and encouraged expression of matrix metalloproteinase (MMP)-2, both key components of muscle regeneration [25]. In that study, it was also demonstrated that activated macrophages cleaved full-length adiponectin into the globular form, helping to stimulate satellite cells via p38 mitogen-activated protein kinase (MAPK) activation and serving as a chemoattractant for further macrophage numbers [25]. An earlier *in vitro* study had demonstrated that the monocyte cell line THP-1 cleaved full-length adiponectin into globular adiponectin whereas Fao hepatocytes, 3T3-L1 adipocytes, and L6 myocytes did not [28], consistent with the work of Fiaschi et al. [25].

Interestingly, recent work using the adiponectin knockout mouse model and adenovirally-mediated adiponectin overexpression was unable to significantly affect skeletal muscle regeneration when compared to wild-type mice [58]. However, (adenovirally-mediated) adiponectin overexpression was capable of improving muscle regeneration in both adiponectin knockout mice and in angiotensin II infused mice (to mimic chronic heart failure condition or aging conditions) [58], suggesting that while adiponectin may not be a primary mediator of skeletal muscle regeneration, its presence or absence can significantly affect the regenerative process. Consistent with this hypothesis, the ability of exercise training to restore regenerative capacity and contractile function in SAMP10 mouse skeletal muscle (a model of accelerated senescence) was nullified when the animals concurrently received adiponectin antibody treatment to lower available circulating adiponectin [56]. Interestingly, the spiny mouse Acomys cahirinus, notable for its exceptional skeletal muscle regenerative capacity, expresses ~2.5-fold greater adiponectin in regenerating muscle compared to that of a C57Bl6 mouse counterpart [70], again suggesting the importance of adiponectin to the regeneration process.

Beyond muscle regeneration, skeletal muscle is also highly adaptable to changes in load bearing (e.g., hypertrophy in response to chronic load bearing; atrophy in response to unloading). Exercise-trained SAMP10 mice demonstrated increased grip strength and muscle mass which as abrogated by anti-adiponectin antibody treatment [56], suggesting adiponectin plays a role in mediating the hypertrophic response to exercise, though it should be noted that endurance exercise was the mode of training in this study. To the best of our knowledge, no study has yet to test the necessity of adiponectin for the hypertrophic response to resistance exercise. Based on these data, it could be speculated that adiponectin is required for hypertrophy, although such speculation is at odds with its role of activating AMPK and therefore suppressing mTOR activity. 

Skeletal muscle expression of adiponectin, its receptors AdipoR1 and R2, and the adaptor protein APPL1 are required to relay the adiponectin signal to the cell interior [71] and the state of load bearing in skeletal muscle dictates the level of expression of these proteins. When overloaded via synergist ablation, mouse soleus fibers increase expression of adiponectin, both adiponectin receptors (AdipoR1 and R2), and APPL1, similar to what occurs in myoblasts as they differentiate and become myotubes *in vitro* [55]. Conversely, after 2 weeks of hindlimb suspension, soleus AdipoR1 expression was reduced, but not adiponectin, AdipoR2, or APPL1. Upon resumption of normal ambulation patterns, soleus AdipoR1, adiponectin, and APPL1 significantly increased [55]. The importance of adiponectin in suppressing muscle atrophy has also been directly demonstrated. Using C2C12 cells, treatment with either globular adiponectin or with glucopyranosyl tetrahydroxydihydroflavonol (GTDF), a mimetic of globular adiponectin, stimulated cell differentiation [57]. Furthermore, GTDF or adiponectin protected against dexamethasone-induced expression of atrogin-1 and MuRF1 (the atrogenes), key genes of the proteolytic pathway which is highly active during muscle atrophy. This effect was consistent in rat gastrocnemius *in vivo* and prevented atrophy [57]. Low expression of adiponectin and elevated expression of the atrogenes was also noted in a study of cachexia in tumour-bearing mice [72]. Thus, muscle expression of adiponectin, its receptors, and associated adapter protein are sensitive to the state of loading and play a role in minimizing proteolysis. We speculate that adiponectin signaling is altered as a mechanism serving to carry out processes related to hypertrophy and atrophy (Figure 1). 

### 3.4. Dystrophy and Inflammation

Adiponectin attenuates inflammatory signaling [73] and has recently been demonstrated to reduce degeneration of muscle in muscular dystrophy. Crossing adiponectin null mice with mdx mice (a murine model of muscular dystrophy), mdx/adiponectin-null mice were generated [74]. Without adiponectin, muscle contractile force was worsened compared to mdx mice, coinciding with higher levels of markers of muscle damage (e.g., plasma creatine kinase, pervading Evans Blue Dye). Restoring adiponectin levels via local gene electrotransfer resulted in reduced markers of inflammation (TNFα, IL-1β, CD68), greater expression of markers of regeneration (Mrf4, myogenin, Myh3, Myh7), and morphological improvements (larger muscle fibers, decreased inflammation and ECM in between fibers). Using adiponectin overexpression in mdx mice, similar improvements (i.e., reduced inflammation, greater expression of myogenic markers, morphological and functional improvements) were observed [75]. Furthermore, treating mdx mice with adiponectin reduced the expression of the Nlrp3 inflammasome, a caspase complex responsible for activating inflammatory cytokines IL-1β and IL-18 [76], providing a potential link between adiponectin and reduced inflammation in skeletal muscle. Importantly, adiponectin treatment of myoblasts isolated from Duchenne Muscular Dystrophy (DMD) patients and cultured into myotubes demonstrated similar results to rodent studies. Analysis of the secretome of DMD-myotubes treated with adiponectin revealed that expression of several inflammatory cytokines (TNFα, IL-17A, and CCL28) was repressed while expression of utrophin was increased [77]. Further, it was recently demonstrated that mesoangioblasts were capable of fusing with dystrophic muscle *in vivo* under the influence of exogenous adiponectin treatment [53]. This is important because treatment of dystrophic muscle with myogenic cells expressing competent dystrophin would ideally result in the replacement of the defective dystrophin gene. If adiponectin can help in these regards, support for adiponectin as an adjunct in novel treatments against muscular dystrophy and associated inflammation is warranted. 

### 3.5. Regulation of Autophagy

Reductions in adiponectin and/or adiponectin signaling could be mediating deleterious effects on skeletal muscle through decreased stimulation of autophagy. Recently, it was demonstrated that insulin resistant L6 skeletal muscle cells have insulin sensitivity restored with adiponectin exposure [78]. Interestingly, this effect of adiponectin was mediated through restoration of autophagy and reduction of ER stress, an effect also captured by rapamycin treatment but lost in Atg5-dominant negative cells that are autophagy-deficient [78]. Activation of autophagy in response to adiponectin (in this case, globular adiponectin) has also been demonstrated in C2C12 cells, promoting myoblast survival and suppressing apoptosis [54]. Furthermore, skeletal muscle from adiponectin KO mice displayed reduced expression of LC3 and beclin-I, key markers of autophagy, as well as histological markers of myopathy (i.e., centrally located nuclei, accentuated fiber cross-sectional area heterogeneity, necrotic fibers) [54]. Interestingly, high fat diet-induced obesity stimulated autophagy, an effect lost in adiponectin-KO mice and restored with adiponectin treatment [79].

### 3.6. Adiponectin Mimetics and Related Proteins

Adiponectin mimetics and related proteins share effects on skeletal muscle similar to those of adiponectin itself. GTDF [57,80] and AdipoRon [67,81] are agonists of the AdipoR and have already been described earlier in this review. Evidence is accumulating that proteins closely related to adiponectin may also play similar roles in skeletal muscle. The C1q/TNF-related protein (CTRP) family has 16 identified family members including adiponectin, many of which form multimeric complexes and have biological functions similar to adiponectin [17]. CTRP3 in particular, is notable due to its positive effect on glucose homeostasis and anti-inflammatory functions [17]. Recently, CTRP3 was demonstrated to be expressed by embryonic skeletal muscle and by differentiating C2C12 myoblasts [82]. Despite being expressed during differentiation, CTRP3 signaling stimulates ERK1/2 activity, promotes proliferation, and delays differentiation of C2C12 myoblasts into myotubes [82]. Thus, it is possible that other members of the CTRP family also play key roles in developing and maintaining a healthy skeletal muscle but have yet to be examined.

## 4. Mechanisms of Benefits of Exercise Mediated by Adiponectin

Unlike most circulating adipokines, adiponectin is inversely associated with adiposity, visceral fat in particular [83]. In general, women express greater plasma adiponectin than men, independent of BMI and fat mass, and there has been suggestion that this relationship is partly influenced by sex hormones [83]. A number of investigations of the sex-related differences in circulating adiponectin throughout adolescence suggest that adiponectin is negatively associated with serum androgens given that there is a drop in adiponectin as young boys progress through puberty, a result not seen to the same extent in young girls and independent of body composition changes during this period [84,85]. 

### 4.1. Acute and Chronic Effects of Exercise on Adiponectin Expression

Circulating adiponectin is negatively associated with insulin resistance, poor glucose control, and diabetes [86,87], and has anti-inflammatory and anti-atherogenic properties [88]. Further, low levels of circulating adiponectin are observed in obese individuals [89,90], those with CVD [91], and some cancers [92]. Consequently, adiponectin has been a prime target for study and manipulation since its initial characterization. Not surprisingly, because physical activity is a potent countermeasure against metabolic and CVD [93,94], studies to determine the relationship between exercise and plasma adiponectin have been plentiful. In rodents, there is evidence to suggest that moderate physical activity (10 weeks voluntary wheel or treadmill running) can increase plasma adiponectin without changes in fat mass [56,95], but this is not clear given that neither 10 weeks of endurance running at 70% maximal running capacity nor 10 weeks of high intensity interval training (HIIT) were shown to significantly increase plasma LMW and HMW adiponectin (as measured by Western Blot) in mice [96]. Similarly, systematic summaries of the relationship between exercise and adiponectin in humans have shown equivocal findings [97,98]. Observations of plasma adiponectin after a single bout of aerobic or resistance exercise reveal small changes, if any, in either direction in acute timelines [90,98,99,100,101], while interventions of repeated bouts of exercise training over weeks or months may cause either an increase [90,98,102], decrease [98,103,104], or no change [98,104,105,106] in this adipokine. This is not unusual when attempting to summarize the results of exercise studies because, much like many of the benefits of an exercise training regime, outcomes are dependent on frequency, mode, intensity, and type of exercise in addition to a host of individual characteristics (e.g., age, health, fitness level, etc.). Similarly, there are challenges in interpreting adiponectin changes in response to exercise because of differences in the sex of study participants, initial body composition, separating fat loss from exercise related changes, and different methods of measuring adiponectin. For example, serum adiponectin was reduced in overweight and obese individuals, but not normal weight middle-age adults following 12 months of aerobic (supervised aquatic exercise for 60 min, twice a week) and resistance training exercise even though all groups improved cardiorespiratory fitness and no group exhibited changes in fat mass following training [104]. In another study, healthy adult men free of any known chronic diseases and grouped according to BMI (i.e., normal BMI versus overweight/obese) and activity level (i.e., sedentary versus active) partook in 2 months of cycle ergometer training (i.e., 3 × 60 min at 50% VO_2_peak) [103]. The study authors measured LMW, MMW, and HMW adiponectin by several ELISA’s and observed reduced total and HMW adiponectin concentrations only after training in the sedentary groups, but not the active groups, regardless of body composition [103]. The findings of these two studies suggest that adiponectin levels in normal weight and/or active adults do not respond to low intensity exercise, whereas overweight/obese individuals show reductions in circulating adiponectin to these exercise intensities, especially when body composition is unchanged. 

In contrast, when exercise is associated with significant body fat loss, it appears that circulating adiponectin is increased. For example, sedentary and obese (30 kg/m^2^ < BMI > 40 kg/m^2^) but otherwise healthy adult (37 ± 7 y) men and women who participated in a supervised aerobic (60 to 75 min/session, three sessions/week at 500 to 600 kcal/session) exercise training regimen and/or reduced calorie diet for 12 weeks, only exhibited changes in adiponectin when the interventions were associated with weight loss [107]. Further, in relatively healthy older (71.2 ± 5.0 years) adult men and women who completed 12 weeks (3 d/wk) of combined moderate intensity endurance (20 min of walking at 60% to 70% of heart rate reserve) and resistance exercise, adiponectin increased over 50% following exercise training [102]. At a similar intensity (45 min at 70% of maximum heart rate, 3×/week) performed by middle-aged hypertensive men, plasma adiponectin was elevated at 8 and 12 weeks of the intervention [108]. Nonetheless, in both studies, increases in circulating adiponectin were either significantly correlated with body fat [102] or occurred in the presence of significant weight loss [108] (Figure 1). Future studies should consider the impact of progressive exercise training on the adipocyte secretome and related molecular signaling, perhaps best achieved with isolated adipocyte studies.

### 4.2. Physical Activity Behaviour and Adiponectin Expression

In contrast to training interventions, large cross-sectional studies of physical activity behaviour and adipokine/inflammatory biomarker expression tend to show a relationship between greater volumes of physical activity and/or moderate to vigorous physical activity (MVPA) and plasma adiponectin that is independent of body fat. For example, older (~60 y) adult women who exhibited greater accelerometer-measured total activity were found to have higher circulating adiponectin, and though this relationship was attenuated after adjusting for BMI, a significant correlation still existed [109]. Moreover, women in the highest quartiles of both total activity and MVPA had significantly higher serum adiponectin than the lowest quartiles (Alessa et al. 2017). This relationship was also observed in young boys and girls (~9 y), where plasma adiponectin was positively associated with VO_2_peak, even though this correlation was weak [110]. In a recent study out of Japan, with one of the largest samples (>10,000) of middle-aged (40 to 69 y) adults, serum concentrations of total adiponectin and HMW adiponectin were greatest in those individuals who were in the highest quartiles of accelerometer-measured light-intensity physical activity (LPA) and MVPA [111]. Not surprisingly, the individuals in the highest quartiles of physical activity also had the lowest BMIs, however the authors used isotemporal substitution analysis to show that replacing 60 min of sedentary time with LPA could be linked to increased total and HMW adiponectin levels by 4% to 13%, respectively, even after adjusting for body fat [111]. In the latter two studies, both girls and women had higher adiponectin levels than boys and men, even though they had lower maximum aerobic capacity [110] or physical activity levels [111], respectively, indicating that the circulating expression of this adipokine is regulated by many factors. Indeed, in middle aged Japanese men and women followed over 3 years, lower plasma adiponectin was observed in individuals who developed type 2 diabetes independent of visceral fat mass even though self-reported physical activity was not different between those with and without diabetes [112]. 

### 4.3. The Link Between Exercise, Adiponectin, and Improved Metabolic Health

Understanding the metabolic signals linked to increased circulating adiponectin could help to explain some of the above observations. However, other than a general idea that adiponectin both regulates [68] and is regulated by plasma FFA [113], the specific trigger initiated by increased physical activity and exercise in humans is not clear [114,115]. It is likely that even this response is multifaceted and, much like many of the observations noted in this review, the data regarding differential processing of the LMW, MMW, and HMW adiponectin are scarce. Interestingly, in one exercise training study, middle-aged adult men and women separated by performance on an oral glucose tolerance test (normal glucose tolerance versus impaired glucose tolerance/non-diabetic) and by presentation with type 2 diabetes performed 20 min of supervised biking or running, 20 min of swimming, and 20 min of cool down sessions, 3 days/week for 4 weeks [116]. In older participants (~50 y) and those with T2D or impaired glucose tolerance, circulating adiponectin was reduced, while following exercise, adiponectin was increased, a result associated with reduced fat mass. These authors also found, however, that muscle adiponectin receptor mRNA was increased following exercise training, and suggested that when translated to receptor protein expression, could be part of the insulin sensitizing effects of regular exercise [116]. Consequently, in addition to investigation into the different molecular weight forms of adiponectin, it would be prudent for exercise studies to examine muscle, liver, and/or other tissue expression of adiponectin receptor expression along with some measure of function. In this context, two recent reports out of the same lab showed that diet, exercise type, and tissue had different interactive effects of the expression of the different molecular weight forms of adiponectin in mice [96,117]. Chronic endurance and HIIT exercise were independently able to attenuate many of the metabolic impairments caused by a high fat diet. Yet, while the expression of LMW and HMW adiponectin in the plasma was relatively unchanged by both exercise types, exercise and high fat feeding interacted to markedly increase muscle HMW adiponectin and reduce adiponectin receptor mRNA versus untrained animals only in muscles suspected to be used during exercise (i.e., the gastrocnemius vs masseter) [96,117]. Further, the addition of a calorically-restricted diet to an endurance exercise program appears to be a potent stimulus to counter the inflammatory and metabolic deregulatory effects of prior high fat feeding, including elevating circulating adiponectin back to normal levels and increasing adiponectin receptor protein expression in responsive tissues, such as the liver [118]. Both the translational and functional implications of these observations remain to be determined, but in the aforementioned studies, the authors noted differential downstream signaling gene products that would indicate altered function of these muscles. 

It is also important to note that although physical exercise benefits several of the processes also influenced by adiponectin, the mechanisms through which exercise mediates these benefits may occur independent of adiponectin expression. Indeed, many of the studies noted above showed some type of advantageous metabolic change regardless of whether circulating adiponectin was increased, decreased, or remained the same. Further, it has been shown that adiponectin KO mice, when exercise trained, demonstrate improvements in expression of mitochondrial markers and activation of intracellular signaling kinases similar to wild type animals, suggesting that adiponectin is not required to mediate exercise-induced benefits in skeletal muscle [119,120]. Nonetheless, it is likely that the physiological change linking exercise to adiponectin expression may or may not occur, but exercise and adiponectin can exert positive metabolic, muscular, and cardiovascular effects independent of each other. 

## 5. Future Directions and Conclusions

The promise of adiponectin as a clinically relevant biomarker and potential therapeutic target continues to expand. Originally deemed an adipose tissue-specific hormone, the past decade has revealed adiponectin expression by numerous tissues including skeletal muscle and the potential for treating not just metabolic diseases but other skeletal muscle conditions such as muscular dystrophy. Its importance for normal physiologic function of skeletal muscle has been demonstrated in studies of muscle development, regeneration, protein turnover, and regulation of inflammatory signaling. The relationship between physical activity (quantity and quality/type) and circulating and local adiponectin isoforms (trimers, hexamers, HMW, and globular) is not yet clear, although a general relationship of high intensity exercise reducing body fat mass leading to greater adiponectin circulation has been established. 

## Figures and Tables

**Figure 1 ijms-20-01528-f001:**
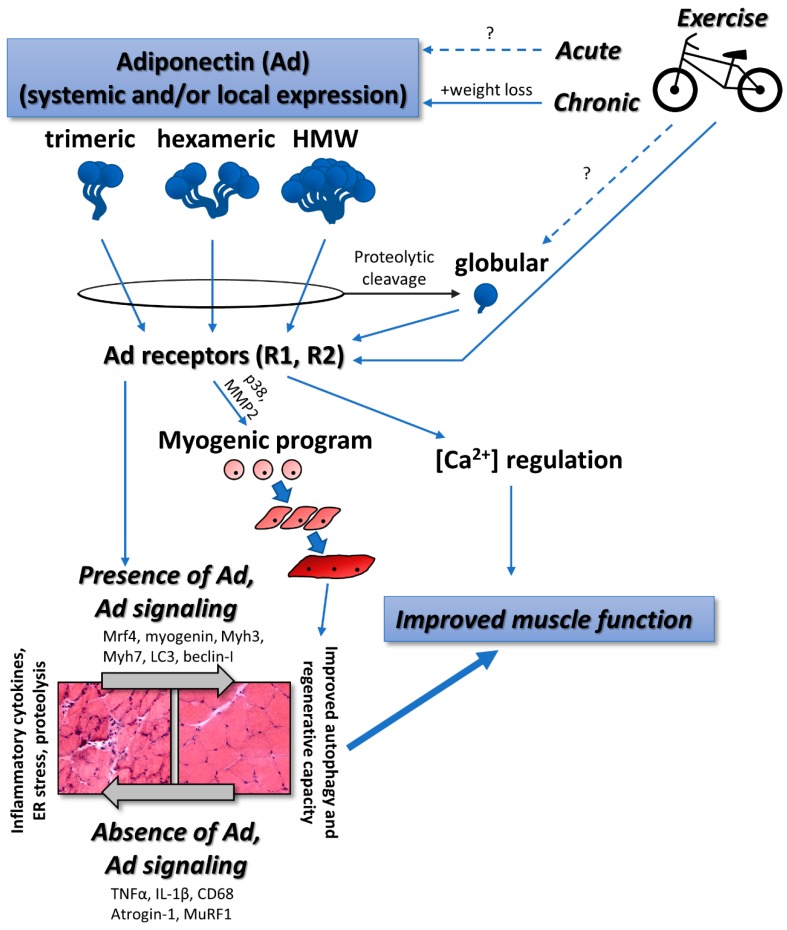
Proposed relationships between adiponectin, exercise, and skeletal muscle function. Multiple isoforms including the proteolytically cleaved globular isoform signal to tissue including skeletal muscle, satellite cells, myoblasts, and differentiated myotubes. Physical exercise generally stimulates increases in adiponectin expression and signaling. Skeletal muscle health is ultimately improved with sufficient adiponectin signaling via improved cellular functions such as autophagy and regeneration and suppression of inflammation, endoplasmic reticulum (ER) stress, and proteolysis. Solid arrows represent relationships, effects, or interactions that are clearly defined in the literature. Broken arrows with “?” represent relationships, effects, or interactions that are not clearly defined in the literature.

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
