# Peer review of "Adiponectin—Consideration for its Role in Skeletal Muscle Health"

_ijms, 2019, doi:10.3390/ijms20071528_

Round 1
Reviewer 1 Report
This is a very well written review, with good literature cited which I believe will be a nice addition to the field.
Minor comments:
Page 5: The muscle growth section might use some re-arranging: development first, then aging. maybe aging in the title of the section since it is emphasized.
Consider the fact that Adiponectin activates AMPK, which increases protein degradation and inhibits mTOR, as a reason for its effects on muscle mass.
Page 7: description of what happens to adiponectin during exercise seems to lack whether Adiponectin is necessary for the hypertrophy response to exercise. Have these experiments been done?
The autophagy section references only 2 papers. Is there more literature than this that could add to this section?
Page: re the association of adiponectin levels to adiposity: maybe suggest that experiments with isolated adipocytes to study the molecular signaling involved in adiponectin secretion are required to elucidate this. The point in the conclusion that this work needs to be done seems excellent. Is adiponectin one of the proteins in the exercise-induced secretome (Hill et al 2018; Whitman et al 2018; Parker et al 2019)? Might be worth considering.
Author Response
The authors would like to thank this reviewer for these insightful comments, all of which have strengthened the paper. Below we have addressed each comment.
Page 5: The muscle growth section might use some re-arranging: development first, then aging. maybe aging in the title of the section since it is emphasized.
Consider the fact that Adiponectin activates AMPK, which increases protein degradation and inhibits mTOR, as a reason for its effects on muscle mass.
This section has been rearranged accordingly with additional statements about AMPK/mTOR signalling. We believe that this greatly improves this section and follows a more logical progression.
Page 7: description of what happens to adiponectin during exercise seems to lack whether Adiponectin is necessary for the hypertrophy response to exercise. Have these experiments been done?
To the best of our knowledge, no. One study using SAMP10 mice demonstrated increased muscle mass and grip strength in response to endurance training, an effect that was abolished by muting adiponectin with antibody treatment. Based on this, it could be hypothesized that adiponectin is required for hypertrophy. Yet this seems to be at odds with its role of activating AMPK and therefore suppressing mTOR activity. We have added some additional text to highlight this issue.
Page: re the association of adiponectin levels to adiposity: maybe suggest that experiments with isolated adipocytes to study the molecular signaling involved in adiponectin secretion are required to elucidate this. The point in the conclusion that this work needs to be done seems excellent. Is adiponectin one of the proteins in the exercise-induced secretome (Hill et al 2018; Whitman et al 2018; Parker et al 2019)? Might be worth considering.
We have added in a statement in the appropriate section regarding examining isolated adipocytes.
A question about the studies you’ve listed here: searching with those author names and combinations of the terms “exercise, secretome, muscle, adipose, adipokine, myokine” are not returning the articles. Could you please expand those citations so that we may investigate further?
Two articles we retrieved that examine exercise-induced changes in the muscle or muscle/adipose secretome do not identify adiponectin as one of the “significant” factors.
Identification of human exercise-induced myokines using secretome analysis. Catoire M, Mensink M, Kalkhoven E, Schrauwen P, Kersten S. Physiol Genomics. 2014 Apr 1;46(7):256-67.
Identification of novel putative adipomyokines by a cross-species annotation of secretomes and expression profiles. Schering L, Hoene M, Kanzleiter T, Jähnert M, Wimmers K, Klaus S, Eckel J, Weigert C, Schürmann A, Maak S, Jonas W, Sell H. Arch Physiol Biochem. 2015;121(5):194-205.
Based on adiponectin’s absence from the two references listed here, we have not added anything into the manuscript. We will wait to see your decision if we should include that information.
Reviewer 2 Report
The authors summarized the role of adiponectin in the skeletal muscle function and maintenance and the effect of exercise on adiponectin signaling. I think that this review article is well organized and informative.
I recommend to add a figure which describes molecular mechanisms regarding skeletal muscle depletion and regeneration and the effect of adiponectin signaling on them.
Author Response
I recommend to add a figure which describes molecular mechanisms regarding skeletal muscle depletion and regeneration and the effect of adiponectin signaling on them.
Thank you for raising this issue. The current figure we provided skimmed by some of those mechansims, so we have now added in additional details. We hope that is a satisfactory resolution.